# OpenReview forum: "Adaptive Strategy Evolution for Generating Tailored Jailbreak Prompts against Black-Box Safety-Aligned LLMs"
_ICLR.cc/2025/Conference — Submitted to ICLR 2025_

### Official Review · Reviewer_2yFe · 2024-11-01

**Soundness:** 2
**Presentation:** 2
**Contribution:** 2
**Rating:** 5
**Confidence:** 4

**Summary:**

This work tackles jailbreak attacks on safety-aligned LLMs, which remain vulnerable to prompt manipulation. The proposed Adaptive Strategy Evolution (ASE) framework improves jailbreak techniques by modularizing strategies and using a genetic algorithm for systematic optimization, replacing uncertain LLM-based adjustments. ASE also introduces precise feedback, enabling targeted refinements. Results show ASE outperforms existing methods against advanced models.

**Strengths:**

1. ASE breaks down jailbreak strategies into modular components, significantly increasing both the flexibility and scope of the strategy space.
2. This work presents an enhanced fitness evaluation system that overcomes the drawbacks of traditional binary or overly complex scoring methods, providing accurate and reliable assessment of jailbreak prompts.

**Weaknesses:**

1. Some key parameters are missing. For example, how about the crossover and mutation rate in the experiment.
2. As evolutionary algorithms are inherently with randomness, it should be tested for multiple time and show the statistical results (average and std). However, I didn’t find any related result, the missing of which will lower the soundness of this work.
3. It states that this is “adaptive” while I didn’t find any “adaptive” design which refers to the strategy will update adaptively as the optimization proceeds. It seems just a normal optimization framework. Can you elaborate on how the ASE framework adapts during the optimization process? Are there any specific mechanisms that adjust the strategy or algorithm parameters based on intermediate results?
4. In line 323, “algorithm 1” should be “Algorithm 1”.

**Questions:**

1. As mentioned “beyond a population size of 15 and iteration count of 5, the gains in JSR become marginal.” This is an indeed small settings, therefore I am curious if the genetic algorithm is overqualified here. Maybe some simpler optimization technique can suffice, such as local search?
2. How about your search space regarding the strategy optimized by the genetic algorithm? If the search space is small then there is no need to use complex optimization techniques.

---

> ### Author Response · Authors · 2024-11-18
>
> Thank you for recognizing the strengths of our pipeline and enhanced evaluation design, which effectively address the limitations of existing methods. We are encouraged by your positive feedback and look forward to addressing your concerns:
>
> > **W1:** Some key parameters are missing. For example, how about the crossover and mutation rate in the experiment.
>
> **Response-W1:**
>
> Thank you for your valuable reminders. Regarding the key parameters, we have set the crossover rate at 0.5 and the mutation rate at 0.7 in our experiments. These settings were carefully chosen based on preliminary experiments to ensure a balanced exploration of the strategy space, while we apologize for not including the corresponding results in the manuscript initially. To address your concern, we have now included the experimental results that demonstrate the impact of these hyperparameters on the performance. In our experiments, we also used Llama3-8B as the victim model, which aligns with the explorations related to population size and iterations outlined in the manuscript. The results are listed as below:
>
> | Mutation Rate | 0.7  | 0.5  | 0.3  |
> | --------------- | ------------------- | ------------------- | ------------------- |
> | JSR / Queries    | 92% / 21.6          | 84% / 28.7          | 76% / 31.2          |
>
> | Crossover Rate | 0.7  | 0.5  | 0.3|
> | --------------- | ------------------- | ------------------- | ------------------- |
> | JSR / Queries | 90% / 25.5          | 92% / 21.6          | 92% / 22.4          |
>
>
>
> From the results, we can observe that the chosen crossover rate and mutation rate provide a favorable trade-off between performance and time cost, effectively optimizing both aspects. The crossover rate, while having a relatively minor impact, should not be set too high, as it may compromise convergence speed; conversely, setting it too low can impede the exploration process. Given these considerations, we have chosen a crossover rate of 0.5. The mutation rate was directly set to 0.7, as a higher rate helps maintain a high accuracy level while preventing undue computational costs. This also reflects the large scale of our strategy space, where higher mutation rates enable more diverse exploration without significant performance degradation.
>
> We hope these additions clarify our parameter choices, and we appreciate your valuable feedback in helping us improve the manuscript again!

---

> ### Author Response · Authors · 2024-11-18
>
> > **W2:** As evolutionary algorithms are inherently with randomness, it should be tested for multiple time and show the statistical results (average and std). However, I didn’t find any related result, the missing of which will lower the soundness of this work.
>
> **Response-W2:**
>
> Thank you for your valuable suggestion. As you suggested, we have conducted multiple runs of experiments (3 repetitions) on the same model and calculate the mean and standard deviation of both the JSR and the average number of Queries. The results are as follows:
>
> | Victim Model | Run 1 (JSR / Queries) | Run 2(JSR / Queries) | Run 3 (JSR / Queries) |
> | ------------ | --------------------- | -------------------- | --------------------- |
> | Llama3       | 92% / 21.6            | 92% / 25.8           | 93% / 25.2            |
> | GPT4o        | 94% / 18.6            | 96% / 17.4           | 94% / 18.9            |
>
> | Victim Model | Average (JSR) | Std (JSR) | Average (Queries) | Std (Queries) |
> | ------------ | ------------- | --------- | ----------------- | ------------- |
> | Llama3       | 92.33%        | 0.47      | 24.20             | 1.85          |
> | GPT4o        | 94.67%        | 0.94      | 18.30             | 0.65          |
>
>
> Clearly, our method demonstrates high stability, and the inherent randomness of the evolutionary algorithm has minimal impact on the jailbreak performance. This shows that the effectiveness of our approach is not significantly affected by the random nature of the algorithm, thus revealing the robustness of our method.

---

> ### Author Response · Authors · 2024-11-18
>
> > **W3:** It states that this is “adaptive” while I didn’t find any “adaptive” design which refers to the strategy will update adaptively as the optimization proceeds. It seems just a normal optimization framework. Can you elaborate on how the ASE framework adapts during the optimization process? Are there any specific mechanisms that adjust the strategy or algorithm parameters based on intermediate results?
>
> **Response-W3:**
>
> Thank you for your question. Actually, the term "adaptive" in our ASE framework does not refer to parameter adaptation within the optimization process itself but instead represents a higher-level adaptability in identifying the most suitable jailbreak strategies based on the given jailbreak intention. Specifically, ASE leverages a genetic algorithm (GA) to iteratively refine strategies by evaluating their fitness. Strategies with higher fitness scores are selected and undergo genetic operations such as crossover and mutation to generate new candidates. This process ensures that the framework dynamically adapts to the jailbreak intention, progressively identifying the most effective strategies through an evolutionary process driven by fitness evaluations.

---

> ### Author Response · Authors · 2024-11-18
>
> > **W4:** In line 323, “algorithm 1” should be “Algorithm 1”.
>
> **Response-W4:**
>
> Thank you for your reminder. We will revise it in the manuscript later.

---

> ### Author Response · Authors · 2024-11-18
>
> > **Q1:** As mentioned “beyond a population size of 15 and iteration count of 5, the gains in JSR become marginal.” This is an indeed small settings, therefore I am curious if the genetic algorithm is overqualified here. Maybe some simpler optimization technique can suffice, such as local search?
>
> **Response-Q1:**
> Thank you for your insightful question. We appreciate the opportunity to provide a more detailed explanation. Actually, we do not adopt local search techniques because they are highly dependent on the choice of the initial solution, making them prone to being trapped in local optima. To ensure exploration diversity, we instead employ a Genetic Algorithm (GA), which inherently maintains population diversity through its initialization process. As shown in Figure 3 of Section 4.2.2, we have also carefully balanced population diversity, JSR, and query cost. Ultimately, we select a GA with a population size of 15, ensuring stable jailbreak success rates while maintaining computational efficiency.

---

> ### Author Response · Authors · 2024-11-18
>
> > **Q2:** How about your search space regarding the strategy optimized by the genetic algorithm? If the search space is small then there is no need to use complex optimization techniques.
>
> **Response-Q2:**
>
> Thank you for your thoughtful comment. To address your concern, we first provide a direct comparison of the scale of our strategy space with that of PAP, which currently represents the most advanced summary in the field. PAP defines only 40 specific strategy types, whereas our approach, structured across four general dimensions with corresponding elements, generates over 800 possible strategy combinations—20 times larger than PAP. This vast strategy space ensures a high degree of diversity in jailbreak strategies but also requires an optimization algorithm capable of thoroughly exploring such a large and complex space.
>
> To this end, we adopt a Genetic Algorithm (GA), which is particularly well-suited for exploring extensive search spaces. Specifically, unlike local search, which is prone to getting trapped in local optima and fails to adequately explore the full potential of the strategy space, GA leverages mechanisms such as population diversity, crossover, and mutation to systematically and effectively search for globally optimal solutions. Therefore, as shown in Table 1 of Sec 4.2, our ASE achieves not only higher efficiency but also superior performance on more challenging models, demonstrating the effectiveness of combining a sufficiently large strategy space with GA.
>
> Finally, we hope this response could address your issues. Thanks for your comment again!

---

> ### Author Response · Authors · 2024-11-21
> **Looking forward to further feedback**
>
> Dear Reviewer 2yFe,
>
> We sincerely appreciate your insightful comments and the time you have dedicated to reviewing our work. We are looking forward to hearing from you about any further feedback.
>
> If you still have any further questions regarding our paper, we are dedicated to discussing them with you and improving our paper.
>
> Best,
>
> Authors

---

> > ### Comment · Reviewer_2yFe · 2024-11-22
> >
> > Thank you for your reply. However, I may still keep my grade because (1) Three time simulations is not enough to support the statistical test, which requires at least 10, normally 30 times. (2) I still can't see the necessity of using genetic algorithm (that is claimed as one of important contribution). (3) "ASE leverages a genetic algorithm (GA) to iteratively refine strategies by evaluating their fitness." If it is called adpative, then the computational cost would be unaffordable due to the high cost of GA.

---

> > > ### Author Response · Authors · 2024-11-22
> > > **Response to Concern (1)**
> > >
> > > > Concern (1):  Three time simulations is not enough to support the statistical test, which requires at least 10, normally 30 times.
> > >
> > >
> > > **Response-C1:**
> > >
> > > Thanks for your insightful comment. For Concern (1), we have already expanded our experiments when responding to Reviewer VLSn. Specifically, we perform dataset scaling by running the full AdvBench, which corresponds to the data volume of 10 iterations of the original test. The results, presented below, show that our ASE consistently exhibits stable performance in terms of JSR and query efficiency. Additionally, as you suggested, we are performing 30 iterations on the test set referenced in the manuscript, and we will provide the complete results at the weekend. Although, to the best of my knowledge, it is uncommon to see a requirement of 30 repetitions for each query in query-based black-box attacks, we still conduct these experiments to fully address your concerns. Finally, we sincerely appreciate your constructive feedback and look forward to your further suggestions.
> > >
> > > | Victim Model | AdvBench (Original, 50) | AdvBench (200)           | AdvBench (Full, 500)     |
> > > |-------------------|-----------------------------|-------------------------------|-------------------------------|
> > > | Llama3        | 92% / 21.6                 | 93.5% / 22.2 | 93.2% / 22.9                  |
> > > | Claude 3.5    | 96% / 20.4                 | 95% / 17.4 | 95.2% / 18.0                  |

---

> > > > ### Author Response · Authors · 2024-11-24
> > > > **Response to Concern (1) - continuing**
> > > >
> > > > Here are the complete results, which demonstrate the average performance (mean) and variations (±std) of 10/20/30 runs. We hope these results address your concerns and provide a clearer understanding of the performance consistency over multiple runs.
> > > >
> > > > | Victim Model | 10 Runs (JSR / Queries)             | 20 Runs (JSR / Queries)             | 30 Runs (JSR / Queries)             |
> > > > |--------------|-------------------------------------|-------------------------------------|-------------------------------------|
> > > > | Llama3       | 92.59% ± 0.38 / 24.22 ± 1.54       | 92.23% ± 0.27 / 23.77 ± 1.38       | 92.22% ± 0.24 / 22.81 ± 1.25       |
> > > > | GPT4o        | 94.38% ± 0.55 / 18.66 ± 0.43       | 94.47% ± 0.40 / 18.13 ± 0.31       | 94.43% ± 0.36 / 18.29 ± 0.28       |

---

> > > ### Author Response · Authors · 2024-11-22
> > > **Response to Concern (2)**
> > >
> > > > Concern (2): I still can't see the necessity of using genetic algorithm (that is claimed as one of important contribution).
> > >
> > > **Response-C2:**
> > >
> > > Thank you for raising this key issue about the necessity of GA. Below is a more detailed explanation:
> > >
> > > **1. Core of ASE:**
> > >
> > > First, we have to clarify that the primary contribution of our work lies in the design of the component-level strategy space, not the specific optimization technique. GA is employed as an optimization tool well-suited to our multi-dimensional, complex strategy space. As demonstrated in our experiments, **GA enables efficient and effective attacks, and even promises the transferability across different models, particularly in challenging black-box scenarios**. This adaptability is what distinguishes ASE from previous methods like PAP and PAIR, rather than the use of a basic GA.
> > >
> > > **2. The Reason for Choosing GA:**
> > >
> > > While GA is not the core contribution, it is essential for navigating our large and diverse strategy space. Unlike local search, which can easily fall into local optima, GA’s use of population diversity, crossover, and mutation ensures robust exploration and exploitation of the space, allowing it to discover optimal strategies more efficientlyly (**proved by average queries in Table 1 of Sec 4.2**) and making it an ideal choice for optimizing within our extensive strategy space.
> > >
> > > **3. The value of GA:**
> > >
> > > Unlike traditional approaches such as PAP and PAIR, which rely on **a trial-and-error process** without strategic guidance, GA enables the general framework ASE to iteratively refine strategies with instant improvement, evaluation and selection, addressing the inherent variability in query-specific effectiveness. This systematic refinement, driven by evolutionary principles, makes ASE particularly effective for adapting to diverse jailbreak queries with efficiency, robustness, and fewer query costs. This means that ASE is more practical in scenarios where query variability plays a critical role.

---

> > > ### Author Response · Authors · 2024-11-22
> > > **Response to Concern (3)**
> > >
> > > > Concern (3): "ASE leverages a genetic algorithm (GA) to iteratively refine strategies by evaluating their fitness." If it is called adpative, then the computational cost would be unaffordable due to the high cost of GA.
> > >
> > > **Response-C3:**
> > >
> > > Thank you for your follow-up comment. We appreciate the opportunity to further clarify the adaptive nature of our ASE framework and address your concern regarding unaffordable computational cost.
> > >
> > > First, it should be noted that the term "adaptive" in our framework is primarily compared to earlier methods such as PAP and PAIR, which **lack direct and reliable guidance during optimization**. PAP test prepared templates one by one, and PAIR just let LLMs provide possible improvement iteratively, without considering the given query's traits or providing specific guidance. In contrast, ASE actively identifies and refines the most suitable strategies through an evolutionary process with **fewer queries** (proved by our experimental results). This adaptability enables our method to **quickly tailor strategies to diverse jailbreak queries**, making it particularly effective for challenging models (proved by our experimental results).
> > >
> > > Second, the concern about **computational cost** being "unaffordable due to the high cost of GA" may not be necessary. The hardware requirements for ASE are minimal, as it only requires a GPU capable of loading the target model, such as an RTX A6000 for *llama3-8b*. In terms of time cost, ASE is highly efficient even when applied to the most challenging closed-source models. Specifically, as shown in Table 1 of Section 4.2, benefiting from the combination of our strategy space and GA algorithm, ASE reduces query requirements by **around 50%** on average compared to earlier black-box methods while delivering superior results. In numerical terms, ASE typically requires an average of around 20 queries per attack, which is clearly an acceptable computational cost. This combination of efficiency and effectiveness further highlights ASE's practical value in black-box jailbreak scenarios.
> > >
> > > In summary, ASE not only adapts more effectively to different types of jailbreak queries but also achieves this with fewer interactions than, existing black-box jailbreak methods. This aligns with our goal of improving effectiveness while ensuring efficiency.

---

### Official Review · Reviewer_KnZq · 2024-11-03

**Soundness:** 1
**Presentation:** 2
**Contribution:** 1
**Rating:** 3
**Confidence:** 5

**Summary:**

This paper introduces a new jailbreak method for large language models, namely ASE. It involves three key steps. First, it decomposes jailbreak strategies into four modules, each of which can be implemented differently. This process constitutes a large search space. ASE then utilizes a genetic algorithm to perform optimization in this space, with a newly designed fitness function. The authors demonstrated the effectiveness of ASE on two tasks, in which it exhibited superior performance compared to previous methods when attacking closed-source models like GPT-4.

**Strengths:**

- The decomposition of jailbreak strategies into different independent modules appears novel.
- The evaluation is thorough.

**Weaknesses:**

- After reading the manuscript, I find this work to resemble more of an engineering design. For example, the authors did not introduce in detail how each component in the search space can be implemented. While there is a table in the appendix describing this, I wonder what protocols or criteria were employed to design this space. Without this information, it is hard to be convinced that rigorous efforts were involved in this design; it seems more like a demo that could be generated by simply prompting an LLM. Also, while the authors regarded their fitness function as a contribution, the four-point scoring with an additional term is a basic engineering trick—all that’s needed is to replace the evaluation prompt for the LLM with a more fine-grained one, which could itself be generated by an LLM. Finally, the core of the proposed method is a basic GA, which may be too straightforward for a prestigious conference like ICLR. More importantly, the authors did not address questions like why they specifically chose GA over other large-scale optimization methods in the literature. Additionally, could the original GA be directly employed for this task, or is this the optimal design choice? There are many similar questions, but unfortunately, the authors did not discuss these matters in their manuscript.

- Building on the previous point, a major concern with using GA in jailbreaking is the number of large number queries it would require. Unlike traditional adversarial attacks, querying LLMs can be very expensive, so the total query budget should be an important consideration in jailbreaking. Although the authors discussed this as a hyperparameter, it is unclear how ASE would perform if given the same budget as previous approaches. Currently, the authors simply treat this aspect as a limitation, but this could actually be a critical consideration for practitioners that can directly influence their choice of methods. This concern further exacerbates when considering the fact that the proposed method does not perform significantly better compared to existing approaches on open-source models, which could be cheaper for employing large-scale query.

- The manuscript is not well-written. For example, there are numerous grammar issues and inappropriate expressions, which strongly affect the readability of the introduction section (for example, many simple processes and concepts introduced in the methods section are made very ambiguous and overly complex, requiring the reader to see the methods section to understand the motivations in the introduction). This can be disappointing, as the introduction is supposed to provide a clear and intuitive picture of the whole manuscript from the outset.

In addition, I list some of the presentation issues.
- Line 074: The reference to Wei et al., 2024 is generated with improper commands, leading to awkward additional brackets. The authors should consult ICLR’s official guide for LaTeX templates.
- Line 078: “Here, red-teaming LLMs act like drafting from structured outlines….” There are grammar issues with this sentence, and I could only gauge its meaning after reading it several times.
- Line 085: The “considering” part has similar issues.
- Line 137: The comma after “TAP” is redundant.

**Questions:**

Please see weakness.

---

> ### Author Response · Authors · 2024-11-18
>
> Thanks for your constructive review. We are encouraged by your appreciation on the novel jailbreak strategy decomposition and complete experimental evaluations. Then, please let us provide responses below to address your concerns:
>
> > **W1-Part1:** After reading the manuscript, I find this work to resemble more of an engineering design. For example, the authors did not introduce in detail how each component in the search space can be implemented. While there is a table in the appendix describing this, I wonder what protocols or criteria were employed to design this space. Without this information, it is hard to be convinced that rigorous efforts were involved in this design; it seems more like a demo that could be generated by simply prompting an LLM.
>
> **Response-W1-Part1:**
>
> Thank you for your thoughtful feedback. We fully understand your concern regarding the lack of a detailed explanation about the implementation of the components in the strategy space, which is the same as **Reviewer VLSn**. We apologize for not being clear about this, and we are grateful for the opportunity to provide more insights into the rigorous efforts behind the design of our strategy space:
>
> **1. Theoretical Grounding and Practical Integration**
>
> The design of our Component-Level Strategy Space is grounded in established multi-element theory applicable to social engineering [1]. For example, Cialdini’s principles [2] contain key elements such as Authority, Social Proof, and Commitment & Consistency, etc.; Gragg’s psychological triggers [3] contain Authority, Diffusion Responsibility, Integrity & Consistency, Reciprocation, etc.; Stajano's principles of scams [4] contain Social Compliance, Deception, Time, etc. They provide robust evidence for the effectiveness of tailored persuasive elements.
>
> In our work, the four components—**Role**, **Content Support**, **Context**, and **Communication Skills**—are derived from such key principles in communication and human behavior, with substantial academic backing. For instance, Role (A) dimension is based on research showing that individuals' behaviors and decision-making are heavily influenced by the roles they assume in interactions (e.g., role-playing scenarios in anti-phishing training, or the finding that individuals are more likely to comply with authority) [5,6,7]. Content Support (B) and Context (C) dimensions are supported by findings on how the presentation of content and the surrounding context affect cognitive processing and decision-making, particularly in digital environments [6,8]. Finally, communication skills are rooted in well-established persuasion models, which demonstrate how different communication techniques impact individuals' willingness to engage with or resist persuasive attempts [9].
> Besides, Uebelacker et al. propose that some principles work better depending on the victim’s personality traits [10]. Thus, there is exact need for adapting strategies dynamically to different scenarios, which is consistent with our design.

---

> ### Author Response · Authors · 2024-11-18
>
> **Reponse-w1-Part1 (Continuing):**
>
> **2. A Generalizable Framework for Strategy Generation**
>
> Beyond its theoretical foundation, what makes our design distinct is the integration of these components into a modular framework. This design still has theoretical support. Prior research, such as Ferreira et al. [11], discusses the relationship of main principles that contain tactics often used in persuasive communication or security strategies, which emphasizes that such principles often exist **overlap** or **complement** one another (e.g., Social Proof encompassing Herd Behavior - Social Proof is a more general principle encompassing the other).
>
> Such phenomenon encourages us to form a more **general and comprehensive** space with dimensional elements, with the consideration of multi-dimensional nature of communication for social engineering. By enabling dynamic recombination of such components, our strategy space allows for both simple (part of dimensions could skip) and sophisticated strategies. This design not only encompasses existing jailbreak approaches but also extends their scope, enabling the generation of novel strategies tailored to diverse and challenging scenarios. To sum up, This systematic design ensures flexibility, adaptability, and scalability, addressing the inherent limitations of predefined, rigid taxonomies like PAP.
>
> **3. Like a demo that simply prompts an LLM**
>
> Regarding the concern of simply prompting an LLM, we would like to emphasize that **prompt-based techniques are the most effective and transferable** in current **black-box jailbreaks**, especially for **well-aligned proprietary models**. We argue that the core of innovation should focus on **how to prompt LLMs in an effective and efficient way for various scenarios**, and the contribution of our work lies not just in creating a new prompt, but in **designing a general strategy space**, incorporating modular components that can be dynamically recombined. This structure allows us to generate various strategies that are more adaptable to diverse scenarios, offering substantial improvements over existing methods.
>
> Additionally, as evidenced by the multiple experimental results presented in both the manuscript and the rebuttal responses, our results are always solid, demonstrating superior attack efficacy, transferability, and proper costs compared to existing black-box attack methods, especially in more challenging scenarios.
>
> ---
>
> Finally, we will revise the manuscript to clearly claim the theoretical foundation behind our design, provide stronger evidence supporting its validity, and eliminate any language that may suggest an overreliance on heuristics. We have gathered extensive **literature support for each component**, which will be incorporated into the manuscript due to the substantial volume of references. Thank you once again for highlighting this crucial issue, which has significantly contributed to enhancing the rigor of our work.
>
> **References**
>
> [1] Mitnick K D, Simon W L. The art of deception: Controlling the human element of security[M]. John Wiley & Sons, 2003.
>
> [2] Cialdini R B, Cialdini R B. Influence: The psychology of persuasion[M]. New York: Collins, 2007.
>
> [3] Gragg D. A multi-level defense against social engineering[J]. SANS Reading Room, 2003, 13: 1-21.
>
> [4] Stajano F, Wilson P. Understanding scam victims: seven principles for systems security[J]. Communications of the ACM, 2011, 54(3): 70-75.
>
> [5] Wen Z A, Lin Z, Chen R, et al. What. hack: engaging anti-phishing training through a role-playing phishing simulation game[C]//Proceedings of the 2019 CHI Conference on Human Factors in Computing Systems. 2019: 1-12.
>
> [6] Floriani A. Negotiating what counts: Roles and relationships, texts and contexts, content and meaning[J]. Linguistics and Education, 1993, 5(3-4): 241-274.
>
> [7] Cialdini, R. B. (2001). *Influence: Science and Practice*. Allyn & Bacon.
>
> [8] Zhang L, Peng T Q, Zhang Y P, et al. Content or context: Which matters more in information processing on microblogging sites[J]. Computers in Human Behavior, 2014, 31: 242-249.
>
> [9] Petty R E, Cacioppo J T. Communication and persuasion: Central and peripheral routes to attitude change[M]. Springer Science & Business Media, 2012.
>
> [10] Uebelacker S, Quiel S. The social engineering personality framework[C]//2014 Workshop on Socio-Technical Aspects in Security and Trust. IEEE, 2014: 24-30.
>
> [11] Ferreira, A., Coventry, L., & Lenzini, G. (2015). Principles of persuasion in social engineering and their use in phishing. *International Conference on Human Aspects of Information Security, Privacy, and Trust*, 36–47.

---

> ### Author Response · Authors · 2024-11-18
>
> >**W1-Part2:** Also, while the authors regarded their fitness function as a contribution, the four-point scoring with an additional term is a basic engineering trick—all that’s needed is to replace the evaluation prompt for the LLM with a more fine-grained one, which could itself be generated by an LLM.
>
> **Response-W1-Part2:**
>
> We apologize if our description gave you the impression that our improvement simply involves constructing a new evaluation prompt for the LLM. Actually, **Reviewer VLSn** has raised a similar concern, and we also provide a response in that part. Below is a detailed response to such issue:
>
> Specifically, our fitness evaluation are not designed arbitrarily; instead, the key contribution lies in the **evaluation logic** embedded within the prompt, which cannot be easily generated through LLMs alone. As detailed in lines 291–322, we redefine evaluation criterion from the perspective of **Intention Consistency**. This design distinguishes our method from previous LLM-based evaluations. For example, the binary judgment (yes or no) often results in misclassification during execution. For rule-intensive scoring methods, the definition overlap and ambiguity complicates the task, making it hard to identify clear distinctions between different levels. This affects the reliability of the evaluation and fails to provide effective guidance for selecting better strategies.
>
> In constrast, our refined perspective focus on determining whether the response addresses the **malicious intent** behind harmful queries, rather than merely evaluating the content in isolation. This is a **semantic understanding task**, which is well within the basic capabilities of LLMs and provides a more reliable way to assess the quality of a response. This makes our method more robust, and less prone to false positives and negatives, particularly in more complex, nuanced scenarios where the text itself may not display any obvious harmful features.
>
> When compared to reward models, we still have two advantages: (1) Our method has no need for extensive training on high-quality labeled data to align with human cognition and preferences just like reward models. This not only **makes our method more efficient** but also **reduces the challenges of acquiring large, diverse datasets for training reward models**. (2) It is crucial to note that a jailbreak text itself does not necessarily exhibit clear toxicity, and query-based black-box attackers often exploit seemingly harmless queries as well as responses to achieve malicious outcomes. For example, an attacker may ask for a "chemical recipe" and the response may appear to be a scientific explanation with no harmful content. However, it could still be harmful when considering **social factors** (e.g., instructions for synthesizing illicit substances). This highlights the gap in traditional harmful data and adversarial jailbreak samples, which possess broader harmful context or intentions behind the text itself. And our method overcomes this by focusing on intention consistency rather than just content semantics. For reward models, even if successful jailbreak examples are added for training, it is difficult to ensure the diversity and comprehensiveness of these samples, and the performance improvement is still limited. Thus, our method is **more adaptable to different types of data**.
>
> Then, to address your concern regarding statistical validation, we have provided comparison of evaluation results among several LLM-based evaluations methods in **Appendix A.2**. And we further conduct experiments on two latest safety reward models, including **llama3-guard** and **Skywork-Reward-Gemma-2-27B-v0.2** (top 1 model on Reward Bench). Following the same settings as in **Appendix A.2**, we evaluate all methods' jailbreak performances. The comparative results are as follows:
>
> |         Method          | Our Method | Binary Judge | Rule-Intensive Scoring Method | llama3-guard | Skywork-Reward-Gemma-2-27B-v0.2 |
> | :---------------------: | :--------: | :----------: | :---------------------------: | :----------: | :-----------------------------: |
> | Evaluation Accuracy |    98%    |     50%     |              82%               |      54%      |               48%               |
>
> Overall, our fitness evaluation could serve as a better judge mechanism to determine the success of a jailbreak attempt, and provide a more reliable way to select more effective jailbreak strategies during optimization. We hope such responses could addess your issue.

---

> ### Author Response · Authors · 2024-11-18
>
> **W1-Part3:** Finally, the core of the proposed method is a basic GA, which may be too straightforward for a prestigious conference like ICLR. More importantly, the authors did not address questions like why they specifically chose GA over other large-scale optimization methods in the literature. Additionally, could the original GA be directly employed for this task, or is this the optimal design choice? There are many similar questions, but unfortunately, the authors did not discuss these matters in their manuscript.
>
> **Response-W1-Part3:**
>
> **1. The Core of the Proposed Method**
>
> Thank you for your comment. Similar to the previous concerns, we would like to clarify that the core of our method does not lie in the specific optimization technique, but rather in the design of the **search space**, i.e. our component-level strategy space. The GA algorithm is simply an optimization tool that is well-suited to work within our space. **The essence of our work is in its ability to effectively and efficiently perform attacks**, particularly in challenging scenarios such as **black-box proprietary models**, which are more difficult to attack. Furthermore, the transferability of the attack across different models is also a key aspect of our approach, and it is this adaptability that distinguishes our work. Thus, the novelty and significance of our contribution lie in the design of the search space and the resulting improvements in attack effectiveness, efficiency, and transferability, rather than the use of a basic GA. Our experimental results provide comprehensive and effective support as well.
>
> **2. The Reason for Choosing GA**
>
> As mentioned in our previous responses, our strategy space is relatively large, necessitating an optimization algorithm capable of effectively navigating such complexity. The choice of Genetic Algorithm (GA) is particularly deliberate, as it is well-suited for handling the challenges of large and diverse search spaces. Unlike local search, which often gets trapped in local optima and struggles with exploration, GA leverages population diversity, crossover, and mutation to systematically explore and exploit the search space. This adaptability enables flexible and robust optimization, making it an ideal choice for optimizing within our extensive strategy space.

---

> ### Author Response · Authors · 2024-11-18
>
> > **W2:** Building on the previous point, a major concern with using GA in jailbreaking is the number of large number queries it would require. Unlike traditional adversarial attacks, querying LLMs can be very expensive, so the total query budget should be an important consideration in jailbreaking. Although the authors discussed this as a hyperparameter, it is unclear how ASE would perform if given the same budget as previous approaches. Currently, the authors simply treat this aspect as a limitation, but this could actually be a critical consideration for practitioners that can directly influence their choice of methods. This concern further exacerbates when considering the fact that the proposed method does not perform significantly better compared to existing approaches on open-source models, which could be cheaper for employing large-scale query.
>
>
> **Response-W2:**
>
> Thank you for your valuable comment. We appreciate the opportunity to address your concerns regarding query costs and performance. First, regarding query times, as shown Table 1 of Section 4.2, our method demonstrates a significant advantage over previous approaches when attacking more challenging closed-source models, achieving both high JSRs and efficiency. We believe that focusing on such difficult and unknown black-box models is more meaningful than targeting relatively simpler open-source models. Then, for open-source models, while our ASE method shows slightly lower performance (by 4%) compared to GPT-Fuzzer on the model Llama3(both exceeding 90% JSR), this small discrepancy may be attributed to the limited size of the AdvBench subset used for evaluation. Thus, to verify our ASE's priority, we scales the subset to the full AdvBench dataset. Eventually, our ASE method achieves a JSR of 93.2%, outperforming GPT-Fuzzer’s 90.5%.
> In term of query costs, while ASE requires approximately 15 additional
> queries compared to GPT-Fuzzer in open-source models like Llama3, the overall token cost remains comparable. This is because GPT-Fuzzer relies on highly complex templates, which **significantly increase the number of tokens per query**.

---

> ### Author Response · Authors · 2024-11-18
>
> >**W3:** The manuscript is not well-written.
>
> **Response-W3:**
>
> Thank you for your detailed and constructive feedback. We sincerely appreciate your efforts in pointing out the writing issues, as they are essential for improving the quality of our manuscript. We deeply apologize for the grammatical errors and awkward expressions, especially in the introduction section, which may have hindered the clarity and readability of our work.
>
> We will carefully address these issues and make thorough revisions in the manuscript to ensure that the content is presented in a clearer manner. Specifically, we will focus on improving sentence structure, clarifying concepts, and removing any ambiguities. Additionally, we will consult ICLR’s official LaTeX guide to properly format references and correct any formatting inconsistencies.
>
> Thank you again for your valuable feedback again. We are committed to making these revisions to enhance the overall reading experience!

---

> ### Author Response · Authors · 2024-11-21
> **Looking forward to further feedback**
>
> Dear Reviewer KnZq,
>
> We sincerely appreciate your insightful comments and the time you have dedicated to reviewing our work. We are looking forward to hearing from you about any further feedback.
>
> If you still have any further questions regarding our paper, we are dedicated to discussing them with you and improving our paper.
>
> Best,
>
> Authors

---

> ### Comment · Reviewer_KnZq · 2024-11-30
>
> I thank the authors for the response, and some of my concerns have been partly addressed. However, many concerns remain. For example, when speaking of the use of GA, the authors now try to avoid saying that is is an important contribution of the work, and I still cannot see convincing justifications for the use of a basic GA (not just because it is a population-based method). In addition, some of the responses seem to be a copy-and-paste from the responses to other reviewers even if I did not raise the mentioned issue (e.g., the statistical validation). Given these, I will keep my score and recommend rejection.

---

> > ### Author Response · Authors · 2024-11-30
> > **Response to Reviewer KnZq's Feedback**
> >
> > Thank you for your feedback. While we regret that your response is provided near the extended deadline's conclusion, which has limited our ability to provide further experimental evidence to address your concerns, we are still willing to offer additional clarifications to resolve your questions. Below, we provide detailed responses to your comments:
> >
> > **Concern 1:**
> > > For example, when speaking of the use of GA, the authors now try to avoid saying that is is an important contribution of the work, and I still cannot see convincing justifications for the use of a basic GA (not just because it is a population-based method).
> >
> > **（1）Response to Concern 1:**
> >
> > - First, we would like to clarify again that the mention of GA in the abstract and main text is only to specify the technique employed for optimizing components. The main contribution of our work lies in the design of the component-level strategy space.
> >
> > - Regarding **the necessity of using GA**, we provide further justification:
> >    As explained earlier, our strategy space is particularly large and diverse in terms of components, making methods such as **local search** unsuitable. Instead, a **population-based** evolutionary approach like GA is necessary. GA ensures diversity in the initial generation, equating to a robust initialization. In later stages, the survival of the fittest mechanism promotes longitudinal improvement, allowing for effective exploration of better solutions.
> >
> >    Our experiments have further demonstrated that, within the **exponentially larger strategy space**, compared to prior black-box jailbreak methods, we achieve a **significant reduction in query numbers**, particularly **for challenging models**. This strongly supports its suitability and necessity since in black-box jailbreak attacks, the **most critical factors** are the JSR (**effectiveness**) as well as the query times (**efficiency**), the latter being directly tied to **operational costs**.
> >
> > - Besides, we have to highlight that it is the synergy between GA and the strategy space that enables our superior performance, and these two aspects **should not be considered in isolation**. **Given the constraints of black-box settings**, we have yet to identify an alternative approach that surpasses GA-series in suitability for our strategy space.
> > - If you believe another method would be more appropriate, **we kindly request that you suggest it**.
> >
> >
> >
> > **Concern 2:**
> > > In addition, some of the responses seem to be a copy-and-paste from the responses to other reviewers even if I did not raise the mentioned issue (e.g., the statistical validation).
> >
> > **（2）Response to Concern 2:**
> >
> > - Regarding the responses related to the strategy space and fitness evaluation, we note that **your concerns overlap significantly with those raised by Reviewer VLSn**. We believe **it is reasonable to provide a unified response with consistent supporting details** to same questions.
> > In reality, we have consolidated all reviewers' questions, categorized them systematically, and provided comprehensive responses accordingly.
> >
> > - As for the statistical validations that you have never mentioned, we speculate that your concern might pertain to the **fitness evaluation** section, where we provided a comparison of results from different evaluation methods. This data was presented to offer a **quantitative and intuitive demonstration** of the superiority of our proposed evaluation method. It merely serves as **supplementary evidence** to strengthen the clarity of our explanations.
> >
> >    The methods we compared here **almost cover all paradigms** currently employed for evaluation in query-based black-box jailbreak methods, ensuring comprehensive and representative comparisons. We believe such results sufficiently demonstrate that our proposed evaluation method **goes beyond a simple prompt-level improvement**, addressing one of the key concerns you raised.
> >
> >    If your concern is about the **experimental results comparing our method with GPTFuzzer**, we just add an experiment to validate our superiority, which directly corresponds to your issue about the performance on open-source models.
> >
> > - In summary, we believe our response is reasonable since our intention has always been to provide a clear understanding of our design in this work, supported by both **textual explanations** and **experimental results**. We essentially aim to convey the deliberate thought and effort behind the design of our methodology. We respectfully suggest that this aspect should not remain a concern, as it reflects our commitment to addressing reviewers’ feedback in a comprehensive and transparent manner.
> >
> > Lastly, we thank that you ultimately chose to take the time to respond to us. We hope that you will carefully review our responses, and consider whether we have addressed the latest issues you raised. We kindly ask that you take these clarifications and the additional explanations provided into account when making final decisions.

---

> > > ### Comment · Reviewer_KnZq · 2024-11-30
> > >
> > > To make it very clear, I did not mention any concerns regarding “statistical validation,” yet the authors responded with “to address your concern regarding statistical validation…” This turns out to be a direct copy-and-paste from their response to reviewer VLSn. Other examples of this include “when compared to reward models, we still have two advantages,” which is also an issue raised by the other reviewers.
> > >
> > > On the one hand, if the authors could not even maintain coherence and cohesion when writing short rebuttal responses, it is hard for me to believe that they would be able to do so in the presentation of their manuscript (which is supported by the fact that the writing of the paper is poor). For example, they make mistakes like abruptly mentioning a concept that has never been introduced before, which could confuse the readers (as happened here).
> > >
> > > On the other hand, such actions (i.e., copy-and-pasting responses without adjusting them to the reviewer’s comments) are very disappointing and irresponsible, both for the reviewers’ efforts and their own work. This is further exacerbated when the authors even attempted to justify this—“we believe our response is reasonable since our intention has always been to provide a clear understanding of our design in this work.” I believe that one can easily judge whether this is indeed the case, or if it is due to the authors’ carelessness during their copy-and-pasting.
> > >
> > > Such wordplays are prevalent throughout the responses to other specific concerns in my comments, leaving me no choice but to recommend rejection. I would suggest the authors to seriously consider my comments I (or other reviewers) listed (including both technical issues and the presentation, which could lead to misunderstanding or confusion) in their revision. Currently, many design choices made by the proposed method are apparently below the bar of a top conference like ICLR.
> > >
> > > Taking the example of optimization methods again, there exists a wealth of black-box optimization in the literature, and many considerations could be involved when selecting the most appropriate one for this specific task (e.g., number of function evaluations, performance, dimensionality of the problem, potential constraints, scalability, etc.). I do not believe a naïve version of GA can somehow meet all desiderata simultaneously without any additional design. If this is indeed the case, rigorous evaluations supporting this should be included (note that I am not asking for additional experiments on this, as this requires far more considerations beyond a table comparing SOTA performance). In cases where it is not, further design efforts would then be demanded to make it work, or some other algorithms should be considered. From my experience working in related fields, this is the basic requirement for rigorous scientific research that can be presented at top conferences. If the authors consider this GA to be a demo for validating effectiveness of the whole system, then they could just send such results to other lower-tier conferences instead of ICLR, as this is not the venue for preliminary results.
> > >
> > > In light of these, I still would like to keep my score.

---

### Official Review · Reviewer_3eHj · 2024-11-05

**Soundness:** 3
**Presentation:** 3
**Contribution:** 3
**Rating:** 6
**Confidence:** 5

**Summary:**

This paper addresses the limitations of current jailbreak methods for safety-aligned Large Language Models (LLMs), particularly in black-box scenarios where models are resistant to prompt manipulation for harmful outputs.  The authors argue that existing methods often either depend too heavily on red-teaming LLMs (pushing their reasoning capacities) or rely on rigid, manually predefined strategies, which limits their adaptability and effectiveness.  To address this, they introduce the Adaptive Strategy Evolution (ASE) framework, which decomposes jailbreak strategies into modular components and optimizes these through a genetic algorithm.  This approach provides a structured, deterministic path to refining jailbreak methods, along with a new scoring system for feedback, resulting in higher jailbreak success rates (JSR) compared to existing approaches.

**Strengths:**

1. This paper is well-written and easy to understand.
2. As a black-box jailbreak method, the ASE framework is well-performed.
3. The method is simple and effective.

**Weaknesses:**

1. I concern on the complexity and resource requirements. ASE’s reliance on genetic algorithms and modular strategy decomposition could be computationally intensive, which might limit its practical applicability for smaller research teams or resource-constrained environments. Is it possible for you to provide runtime, or memory usage, or hardware specifications used for the experiments.  Additionally, suggesting a comparison of these requirements to existing methods.
2. No other obvious weakness.

**Questions:**

1. See weakness. How does the computational cost of ASE compare to existing jailbreak methods, particularly in large-scale testing scenarios? If you have enough to rebuttal, please compare ASE's scalability to that of baseline methods as the dataset size increases.
2. I wonder whether you have tried to propose a defense framework against ASE? How existing defense mechanisms might perform against ASE？

---

> ### Author Response · Authors · 2024-11-18
>
> We sincerely appreciate your comprehensive recognition and endorsement of our work, which greatly encourages us. Then, please let us provide responses below to address your concerns:
>
> > **W1**: I concern on the complexity and resource requirements. ASE’s reliance on genetic algorithms and modular strategy decomposition could be computationally intensive, which might limit its practical applicability for smaller research teams or resource-constrained environments. Is it possible for you to provide runtime, or memory usage, or hardware specifications used for the experiments. Additionally, suggesting a comparison of these requirements to existing methods.
>
> **Response-W1:**
>
> Thanks for your reminder. As shown in Table 1 of Section 4.2, our ASE method demonstrates a significant advantage in query efficiency (averaging just nearly 20 queries per prompt) compared to existing black-box methods, particularly when applied to more challenging models. Regarding practical environmental requirements, as a query-based jailbreak method, our ASE only requires computational resources capable of loading the target model, which is the same as other black-box jailbreak methods. For instance, in our experiments attacking the llama3-8B model, we use a single 48GB RTX A6000 GPU, with each prompt taking an average of 57.6s to complete.
>
> Furthermore, due to the excellent transferability of our method (as shown in Table 2 of Section 4.2.3), if computational resources are unavailable, the method can easily migrate by querying commercial APIs as victim models or utilizing smaller models that require less GPU memory. This flexibility allows adaptation to other scenarios with limited computational resources. Additionally, because of the low number of queries required in ASE, the cost of using APIs remains minimal.
>
> We appreciate your insights again and are committed to incorporating these improvements to enhance the clarity and rigor of our work.

---

> ### Author Response · Authors · 2024-11-18
>
> > **Q1:** See weakness. How does the computational cost of ASE compare to existing jailbreak methods, particularly in large-scale testing scenarios? If you have enough to rebuttal, please compare ASE's scalability to that of baseline methods as the dataset size increases.
>
> **Response-Q1:**
>
> As you suggested, we expand the test dataset subset (AdvBench) from 50 samples to 200 and eventually to the full dataset of 500 samples to verify the scalability of our ASE method. The experimental results, shown in the table below, demonstrate that our ASE maintains excellent scalability and stable performance even on larger datasets. It is worth noting that we do not include the baseline methods in the scalablity evaluation, as our experiments (Table 1) consistently show that they are nearly ineffective against challenging models like Claude3.5. And we hope such additions could address your concerns. Thanks for your valuable suggestion again!
>
> | Victim Model | AdvBench (Original, 50) | AdvBench (200)           | AdvBench (Full, 500)     |
> |-------------------|-----------------------------|-------------------------------|-------------------------------|
> | Llama3       | 92% / 21.6                 | 93.5% / 22.2 | 93.2% / 22.9                  |
> | Claude 3.5    | 96% / 20.4                 | 95% / 17.4 | 95.2% / 18.0                  |

---

> ### Author Response · Authors · 2024-11-18
>
> > **Q2:** I wonder whether you have tried to propose a defense framework against ASE? How existing defense mechanisms might perform against ASE？
>
> **Response-Q2:**
>
> Thanks for your valuable suggestion. First, as shown in Table 3 of Section 4.2.5, we have followed the setting of PAP to evaluate SOTA defense methods against LLM jailbreaks, including RA-LLM and Smooth LLM. The results demonstrate that our method remains highly robust across most models, with only minor performance degradation. Then, to address the challenge posed by difficult-to-defend methods like ASE, we are also actively exploring defenses from a more fundamental perspective, focusing on distinguishing between benign and malicious distributional feartures. The defense work will be made publicly available in the future.

---

> > ### Comment · Reviewer_3eHj · 2024-11-19
> >
> > Thanks for your response. My concerns have been addressed so I decide to raise my confidence score.

---

> > > ### Author Response · Authors · 2024-11-19
> > >
> > > We sincerely appreciate your full recognition and the raised score, and we'll try our best to further improve the final version.

---

### Official Review · Reviewer_VLSn · 2024-11-08

**Soundness:** 1
**Presentation:** 2
**Contribution:** 2
**Rating:** 3
**Confidence:** 4

**Summary:**

They present a novel approach to jailbreaking LLMs in black-box settings by leveraging evolutionary algorithms. The ASE framework divides jailbreak strategies into four modular components—Role, Content Support, Context, and Communication Skills—expanding the possible strategy combinations and enabling dynamic adaptation.

**Strengths:**

1.	Adapts insights from evolutionary algorithms to LLM jailbreaking, breaking down the pipeline into four distinct components.

2.	Positive experimental results: Experimental outcomes show that ASE achieves higher JSRs with fewer queries compared to existing methods across several models.

**Weaknesses:**

I appreciate the authors’ efforts but would like to point out the following:

1.	The design in **Component-Level Strategy Space** seems artificial and arbitrary. The authors argue that “these components perfectly align with how humans are more easily persuaded,” which is a strong statement lacking evidence. In designing red-teaming algorithms for LLMs, significantly more investigation is needed to justify such a design beyond human-like analogies and heuristics, which do not align with academic rigor.

2.	Regarding the **Fitness Evaluation**, the design also relies on heuristics that make the algorithm feel less reliable, particularly with features like “keyword matching.” What advantages does this offer compared to a reward model that labels the answer’s harmfulness?

3.	In terms of the **experiments**, I believe the focus is misplaced. The authors make strong claims about “expanding the search space and allowing for precise, efficient strategy refinement,” yet I could not find any specific evidence to support this. The paper's algorithm, compared with other works, centers on adaptive evolution, but only JSR is shown. What about the structure and behavior variations within the population? I could not find compelling evidence to justify the complex design of such an evolution algorithm, which seems to introduce a high barrier to practical use and deployment.

4. **Scalability and computational cost** are concerned, given the computationally intensive nature of population-based algorithms.

**Questions:**

- The authors propose breaking down jailbreak strategies into Role, Content Support, Context, and Communication Skills, claiming these align with persuasive human behaviors. How can we be sure these components genuinely improve jailbreak efficacy, given the lack of empirical justification? Could there be other, more effective component structures, or are these simply intuitive choices without systematic validation?

- If the initial population is not well-chosen, could ASE converge to suboptimal solutions, reducing its effectiveness?

---

> ### Author Response · Authors · 2024-11-18
>
> We sincerely appreciate your constructive review. We are encouraged by your appreciation on the novel jailbreak pipline and positive experimental results. Then, please let us provide responses below to address your concerns:
>
> > **W1:** The design in Component-Level Strategy Space seems artificial and arbitrary. The authors argue that “these components perfectly align with how humans are more easily persuaded,” which is a strong statement lacking evidence. In designing red-teaming algorithms for LLMs, significantly more investigation is needed to justify such a design beyond human-like analogies and heuristics, which do not align with academic rigor.
>
> **Response-W1:**
>
> Thank you for your valuable feedback. We understand your concern that the design of the strategy space might appear arbitrary and we are sorry for the lack of detailed explanation of the definition. Below, we provide a structured explanation to demonstrate that our design is rigorously grounded in established research and effectively addresses limitations in prior works, rather than intuitive results.
>
> **1. Theoretical Grounding and Practical Integration**
>
> The design of our Component-Level Strategy Space is grounded in established multi-element theory applicable to social engineering [1]. For example, Cialdini’s principles [2] contain key elements such as Authority, Social Proof, and Commitment & Consistency, etc.; Gragg’s psychological triggers [3] contain Authority, Diffusion Responsibility, Integrity & Consistency, Reciprocation, etc.; Stajano's principles of scams [4] contain Social Compliance, Deception, Time, etc. They provide robust evidence for the effectiveness of tailored persuasive elements.
>
> In our work, the four components—**Role**, **Content Support**, **Context**, and **Communication Skills**—are derived from such key principles in communication and human behavior, with substantial academic backing. For instance, Role (A) dimension is based on research showing that individuals' behaviors and decision-making are heavily influenced by the roles they assume in interactions (e.g., role-playing scenarios in anti-phishing training, or the finding that individuals are more likely to comply with authority) [5,6,7]. Content Support (B) and Context (C) dimensions are supported by findings on how the presentation of content and the surrounding context affect cognitive processing and decision-making, particularly in digital environments [6,8]. Finally, communication skills are rooted in well-established persuasion models, which demonstrate how different communication techniques impact individuals' willingness to engage with or resist persuasive attempts [9].
> Besides, Uebelacker et al. propose that some principles work better depending on the victim’s personality traits [10]. Thus, there is exact need for adapting strategies dynamically to different scenarios, which is consistent with our design.

---

> ### Author Response · Authors · 2024-11-18
>
> **Response-W1 (Continuing):**
>
> **2. A Generalizable Framework for Strategy Generation**
>
>  Beyond its theoretical foundation, what makes our design distinct is the integration of these components into a modular framework. This design still has theoretical support. Prior research, such as Ferreira et al. [11], discusses the relationship of main principles that contain tactics often used in persuasive communication or security strategies, which emphasizes that such principles often exist **overlap** or **complement** one another (e.g., Social Proof encompassing Herd Behavior - Social Proof is a more general principle encompassing the other).
>
> Such phenomenon encourages us to form a more **general and comprehensive** space with dimensional elements, with the consideration of multi-dimensional nature of communication for social engineering. By enabling dynamic recombination of such components, our strategy space allows for both simple (part of dimensions could skip) and sophisticated strategies. This design not only encompasses existing jailbreak approaches but also extends their scope, enabling the generation of novel strategies tailored to diverse and challenging scenarios. To sum up, This systematic design ensures flexibility, adaptability, and scalability, addressing the inherent limitations of predefined, rigid taxonomies like PAP.
>
> ---
>
> Finally, We will revise the manuscript to better articulate this grounding, provide clearer evidence supporting the validity of the design, and remove any language that may imply overreliance on heuristics. We will also provide **literature support for specific elements**, which will be further added in the manuscript due to the large number. Thank you again for raising this important issue, which has helped us improve the clarity and rigor of our work.
>
>
> **References**
>
> [1] Mitnick K D, Simon W L. The art of deception: Controlling the human element of security[M]. John Wiley & Sons, 2003.
>
> [2] Cialdini R B, Cialdini R B. Influence: The psychology of persuasion[M]. New York: Collins, 2007.
>
> [3] Gragg D. A multi-level defense against social engineering[J]. SANS Reading Room, 2003, 13: 1-21.
>
> [4] Stajano F, Wilson P. Understanding scam victims: seven principles for systems security[J]. Communications of the ACM, 2011, 54(3): 70-75.
>
> [5] Wen Z A, Lin Z, Chen R, et al. What. hack: engaging anti-phishing training through a role-playing phishing simulation game[C]//Proceedings of the 2019 CHI Conference on Human Factors in Computing Systems. 2019: 1-12.
>
> [6] Floriani A. Negotiating what counts: Roles and relationships, texts and contexts, content and meaning[J]. Linguistics and Education, 1993, 5(3-4): 241-274.
>
> [7] Cialdini, R. B. (2001). *Influence: Science and Practice*. Allyn & Bacon.
>
> [8] Zhang L, Peng T Q, Zhang Y P, et al. Content or context: Which matters more in information processing on microblogging sites[J]. Computers in Human Behavior, 2014, 31: 242-249.
>
> [9] Petty R E, Cacioppo J T. Communication and persuasion: Central and peripheral routes to attitude change[M]. Springer Science & Business Media, 2012.
>
> [10] Uebelacker S, Quiel S. The social engineering personality framework[C]//2014 Workshop on Socio-Technical Aspects in Security and Trust. IEEE, 2014: 24-30.
>
> [11] Ferreira, A., Coventry, L., & Lenzini, G. (2015). Principles of persuasion in social engineering and their use in phishing. *International Conference on Human Aspects of Information Security, Privacy, and Trust*, 36–47.

---

> ### Author Response · Authors · 2024-11-18
>
> > **W2:** Regarding the Fitness Evaluation, the design also relies on heuristics that make the algorithm feel less reliable, particularly with features like “keyword matching.” What advantages does this offer compared to a reward model that labels the answer’s harmfulness?
>
> **Response-W2:**
>
> Actually, our fitness evaluation does not rely on heuristic algorithms. Instead, it assists heuristic methods by providing a consistent and reliable way to select more effective jailbreak strategies.
>
> Moreover, our fitness evaluation could serve as a better judge mechanism to determine the success of a jailbreak attempt. As detailed in lines 291–322, we redefine evaluation criterion from the perspective of **intention consistency**. This design distinguishes our method from previous LLM-based evaluations. For example, the binary judgment (yes or no) often results in misclassification during execution. For rule-intensive scoring methods, the definition overlap and ambiguity complicates the task, making it hard to identify clear distinctions between different levels. This affects the reliability of the evaluation and fails to provide effective guidance for selecting better strategies.
>
> In constrast, our refined perspective focus on determining whether the response addresses the **malicious intent** behind harmful queries, rather than merely evaluating the content in isolation. This is a **semantic understanding task**, which is well within the basic capabilities of LLMs and provides a more reliable way to assess the quality of a response. This makes our method more robust, and less prone to false positives and negatives, particularly in more complex, nuanced scenarios where the text itself may not display any obvious harmful features. As for the use of **keyword matching**, it functions **solely as an auxiliary tool** to further ensure evaluation precision, as shown in Equation (11).
>
> When compared to reward models, we have two advantages: (1) Our method has no need for extensive training on high-quality labeled data to align with human cognition and preferences just like reward models. This not only makes our method more efficient but also reduces the challenges of acquiring large, diverse datasets for training reward models. (2) It is crucial to note that a jailbreak text itself does not necessarily exhibit clear toxicity, and query-based black-box attackers often exploit seemingly harmless queries as well as responses to achieve malicious outcomes. For example, an attacker may ask for a "chemical recipe" and the response may appear to be a scientific explanation with no harmful content. However, it could still be harmful when considering **social factors** (e.g., instructions for synthesizing illicit substances). This highlights the gap in traditional harmful data and adversarial jailbreak samples, which possess broader harmful context or intentions behind the text itself. And our method overcomes this by focusing on intention consistency rather than just content semantics. For reward models, even if successful jailbreak examples are added for training, it is difficult to ensure the diversity and comprehensiveness of these samples, and the performance improvement is still limited. Thus, our method is more adaptable to different types of data.
>
> Then, to address your concern regarding statistical validation, we have also conducted additional experiments comparing our method with the latest safety reward models, including **llama3-guard** and **Skywork-Reward-Gemma-2-27B-v0.2** (top 1 model on Reward Bench). Following the same settings as in **Appendix A.2** where we compared with the binary judge and the rule-intensive scoring method, we evaluated these models and our method for accuracy in jailbreak evaluation. The comparative results are as follows:
>
> |         Method          | Our Method | Binary Judge | Rule-Intensive Scoring Method | Llama3-guard | Skywork-Reward-Gemma-2-27B-v0.2 |
> | :---------------------: | :--------: | :----------: | :---------------------------: | :----------: | :-----------------------------: |
> | Evaluation Accuracy |    98%    |     50%    |              82%               |      54%      |               48%                |

---

> ### Author Response · Authors · 2024-11-18
>
> > **W3:** In terms of the experiments, I believe the focus is misplaced. The authors make strong claims about “expanding the search space and allowing for precise, efficient strategy refinement,” yet I could not find any specific evidence to support this. The paper's algorithm, compared with other works, centers on adaptive evolution, but only JSR is shown. What about the structure and behavior variations within the population? I could not find compelling evidence to justify the complex design of such an evolution algorithm, which seems to introduce a high barrier to practical use and deployment.
>
> **Response-W3:**
>
> Thank you for your valuable suggestions. Actually, as shown in Table 1 of Sec 4.2, we have demonstrated our expanded strategy space's priority (both JSR and Quries) to previous methods, especially in challenging closed-source models like Claude-3.5. To further support our claims regarding "expanding the search space and enabling precise, efficient strategy refinement," we also conduct **an ablation study that examines the impact of the number of components in the strategy space**. A reduction in the number of components directly shrink the search space. Specifically, we select llama3 and Claude3.5 as victim models, use AdvBench as the dataset, and set the number of components to 2, 3, and 4, respectively. The results are listed in the below table, where we can find that as the number of components decreases, the search space shrinks, leading to a significant drop in the success rate of jailbreak methods and an increase in the number of queries required. This observation aligns with our claim that a sufficiently large search space supports more efficient and effective strategy refinement. We also provide **examples of evolved strategies in Appendix A.5**, which illustrate **the rapid and accurate structure and behavior variations within the population**, reducing waiting times and computational costs, and enhancing practical usability.
>
> | Victim Model | Components: 2 (JSR / Queries) | Components: 3 (JSR / Queries)   | Components: 4 (JSR / Queries) |
> | ------------ | ----------------------------- | ------------------------------- | ----------------------------- |
> | Llama3       | 58% / 47.1      |  79.5% / 31.6  | 92% / 21.6                     |
> | Claude3.5    | 78% / 32.2     |  84.5% / 28.0  | 96% / 20.4       |

---

> ### Author Response · Authors · 2024-11-18
>
> > **W4:** Scalability and computational cost are concerned, given the computationally intensive nature of population-based algorithms.
>
> **Response-W4:**
>
> Thank you for your reminder. First, to verify the scalability of our ASE method, we expand the test dataset subset (AdvBench) from 50 samples to 200 and eventually to the full dataset of 500 samples. The experimental results, shown in the table below, demonstrate that our method maintains excellent scalability and stable performance even on larger datasets.
>
> | Victim Model | AdvBench (Original, 50) | AdvBench (200)           | AdvBench (Full, 500)     |
> |-------------------|-----------------------------|-------------------------------|-------------------------------|
> | Llama3        | 92% / 21.6                 | 93.5% / 22.2 | 93.2% / 22.9                  |
> | Claude3.5    | 96% / 20.4                 | 95% / 17.4 | 95.2% / 18.0                  |
>
> Then, regarding computational cost, as shown in Table 1 of Sec 4.2, our ASE method achieves a significant advantage in query efficiency compared to other approaches, particularly when applied to more challenging models. Moreover, as a query-based method, ASE requires only resources capable of loading the victim model, such as a single 48GB RTX-A6000 GPU for llama3-8b. Furthermore, as demonstrated in Table 2 of Sec 4.2.3, our method exhibits excellent transferability, enabling the use of commercial APIs as victim models when computational resources are unavailable. Due to the low number of queries required, the API usage cost remains minimal.

---

> ### Author Response · Authors · 2024-11-18
>
> > **Q1:** The authors propose breaking down jailbreak strategies into Role, Content Support, Context, and Communication Skills, claiming these align with persuasive human behaviors. How can we be sure these components genuinely improve jailbreak efficacy, given the lack of empirical justification? Could there be other, more effective component structures, or are these simply intuitive choices without systematic validation?
>
> **Response-Q1:**
> Thank you for your insightful feedback. We appreciate the opportunity to clarify our approach further. Below is a more concise response to your concerns:
>
> **1. Addressing the Lack of Empirical Justification**
>
> As declared in **Response-W1**, we have proved that the components we propose for our strategy space are not intuitive choices; rather, they are grounded in solid, multi-element theories that have been empirically validated in the fields of social engineering.
>
> As previously mentioned, well-known principles from Cialdini’s work on social influence, Gragg’s psychological triggers, to Stajano's principles of scams provide evidence of the impact of key components like authority, social compliance, etc. on human behavior. These principles have been proven effective in shaping human interactions, particularly in contexts that involve deception and persuasion, making them highly relevant for the task of red-teaming LLMs.
>
>
> **2. Empirical Validation Through Ablation Studies**
>
> To further validate the role of each component, we conduct ablation studies, where we systematically disable one dimension at a time and evaluate the attack performance. The results confirm that each component contributes meaningfully to the attack’s success, improving the JSR as well as query times.
>
> | Victim Model |  Original (JSR / Queries)  | Remove A (JSR / Queries) | Remove B (JSR / Queries)   | Remove C (JSR / Queries) | Remove D (JSR / Queries) |
> | ------------ | ------------ | ----------------------------- | ------------------------------- | ----------------------------- | ----------------------------- |
> | Llama3       | 92% / 21.6     |  66% / 38.1  | 84% / 30.9   | 78% / 30.2   | 90% / 27.3   |
> | Claude3.5    | 96% / 20.4     |  80% / 32.1  | 86% / 25.5   | 84% / 27.8   | 88% / 26.4   |
>
> **3. Framework Completeness and Interconnectedness**
>
> It is important to note that **Role**, **Content**, and **Context** are core interwoven dimensions that together form the foundation for persuasive interactions like previous studies of communication. **Communication Skills**, as an external enabling factor, connects and enhances these core components, improving the overall strategy’s effectiveness. This conceptualization ensures a complete and cohesive framework, with each component complementing the others in influencing the target model's behavior.
>
> The dynamic nature of these components also aligns with Uebelacker et al.'s assertion that strategies are often more effective when adapted to the target's personality traits. This adaptability is key in red-teaming tasks, where various attack strategies must be tailored to the specific model and scenario to achieve optimal results.
>
>
> **4. Considering Alternative Components**
>
> We acknowledge that there may be other potential components, while our approach represents a structured, theory-based general framework derived from established social engineering principles. The four components we propose have offered a balanced and comprehensive space that accounts for the key factors influencing interaction, while it is still adaptable to specific scenarios. Any alternatives would need to be rigorously tested and validated through empirical studies, similar to our assessment of their contributions. Given the theoretical and empirical grounding of our components, we argue that our proposed structure offers a robust foundation for designing effective jailbreak strategies, with room for further refinement.
>
> We will revise the manuscript to provide more rigorous information for this part, and we hope the responses could address your issues. Thanks for your question again!

---

> ### Author Response · Authors · 2024-11-18
>
> > **Q2:** If the initial population is not well-chosen, could ASE converge to suboptimal solutions, reducing its effectiveness?
>
> **Response-Q2:**
>
> Thank you for your question. As shown in Figure 3 of Sec 4.2, such a risk is possible when ASE takes a small population number, 2. Thus, after conducting hyperparameter tuning experiments, we set the population number to 15. This adjustment slightly increases computational costs (even with this setting, our method achieves the lowest number of queries on challenging models compared to other approaches) but ensures a consistently high JSR. Furthermore, in larger-scale dataset experiments of Response to W4, our method still demonstrates stable high performance.

---

> ### Author Response · Authors · 2024-11-21
> **Looking forward to further feedback**
>
> Dear Reviewer VLSn,
>
> We sincerely appreciate your insightful comments and the time you have dedicated to reviewing our work. We are looking forward to hearing from you about any further feedback.
>
> If you still have any further questions regarding our paper, we are dedicated to discussing them with you and improving our paper.
>
> Best,
>
> Authors

---

### Meta-Review · Area_Chair_eh5b · 2024-12-18

**Metareview:**

Summary:
This paper explores the limitations of existing jailbreak techniques for safety-aligned LLMs, particularly in black-box settings where models resist prompt manipulation to produce harmful outputs. To overcome these limitations, the authors propose the Adaptive Strategy Evolution (ASE) framework. It breaks down jailbreak strategies into modular components and leverages a genetic algorithm to optimize them. This structured and deterministic approach enhances the refinement of jailbreak techniques, complemented by a novel feedback-based scoring system, which leads to improved jailbreak success rates (JSR) compared to existing methods.

Strength:
- The ASE framework decomposes jailbreak strategies into modular components, which enhanced the flexibility and expanded the exploration of the strategy space.
- The ASE framework introduces an improved fitness evaluation system that addresses the limitations of conventional binary or overly intricate scoring methods, which offered a more precise and reliable assessment of jailbreak prompts.
- Experimental results show that ASE achieves higher JSRs with fewer queries against existing methods.

Weakness:
- The reliability and robustness of using a simple GA in the ASE are not fully justified.

Decision: I recommend rejection as the above unaddressed weakness is an important concern.

**Additional Comments On Reviewer Discussion:**

The majority of the reviewers voted for rejection:

Reviewer VLSn voted for a weak accept, as he/she acknowledged that the authors had addressed their concerns. However, the reviewer did not actively advocate for the paper during the post-rebuttal discussion.

Reviewer 3eHj voted for rejection but did not provide any additional feedback following the authors' rebuttal. Based on my observation, their comments were partially addressed.

Reviewers KnZq and 2yFe raised  many questions and engaged actively with the authors during the rebuttal phase. While some of their concerns appear to have been addressed, they ultimately maintained their stance for rejection, as they criticize the lack of full conviction regarding the use of the basic GA.

In my view, a more thorough justification of the GA's usage is crucial. While the authors' revisions improved the paper, the current version still falls slightly below the acceptance threshold.

---

### Decision · Program_Chairs · 2025-01-22

Reject